# Spatial distribution and geographical heterogeneity factors associated with poor consumption of foods rich in vitamin A among children age 6–23 months in Ethiopia: Geographical weighted regression analysis

**Sofonyas Abebaw Tiruneh**[1☺]*, **Dawit Tefera Fentie**[2☺], **Seblewongel Tigabu Yigizaw**[2‡], **Asnakew Asmamaw Abebe**[2‡], **Kassahun Alemu Gelaye**[2‡]

**1** Department of Public Health, College of Health Sciences, Debre Tabor University, Debre Tabor, Ethiopia, **2** Department of Epidemiology and Biostatistics, Institute of Public Health, College of Medicine and Health Sciences, University of Gondar, Gondar, Ethiopia

☺ These authors contributed equally to this work.
‡ These authors also contributed equally to this work
* zephab2@gmail.com

## Abstract

### Introduction

Vitamin A deficiency is a major public health problem in poor societies. Dietary consumption of foods rich in vitamin A was low in Ethiopia. This study aimed to assess the spatial distribution and spatial determinants of dietary consumption of foods rich in vitamin A among children aged 6–23 months in Ethiopia.

### Methods

Ethiopian 2016 demographic and health survey dataset using a total of 3055 children were used to conduct this study. The data were cleaned and weighed by STATA version 14.1 software and Microsoft Excel. Children who consumed foods rich in vitamin A (Egg, Meat, Vegetables, Green leafy vegetables, Fruits, Organ meat, and Fish) at least one food item in the last 24 hours were declared as good consumption. The Bernoulli model was fitted using Kuldorff's SaTScan version 9.6 software. ArcGIS version 10.7 software was used to visualize spatial distributions for poor consumption of foods rich in vitamin A. Geographical weighted regression analysis was employed using MGWR version 2.0 software. A P-value of less than 0.05 was used to declare statistically significant predictors spatially.

### Results

Overall, 62% (95% CI: 60.56–64.00) of children aged 6–23 months had poor consumption of foods rich in vitamin A in Ethiopia. Poor consumption of foods rich in vitamin A highly clustered in Afar, eastern Tigray, southeast Amhara, and the eastern Somali region of Ethiopia. Spatial scan statistics identified 142 primary spatial clusters located in Afar, the eastern part

**Data Availability Statement:** All relevant data are within the paper and/or Supporting Information files.

**Funding:** The author(s) received no specific funding for this work.

**Competing interests:** The authors have declared that no competing interests exist.

of Tigray, most of Amhara and some part of the Oromia Regional State of Ethiopia. Children living in the primary cluster were 46% more likely vulnerable to poor consumption of foods rich in vitamin A than those living outside the window (RR = 1.46, LLR = 83.78, P < 0.001). Poor wealth status of the household, rural residence and living tropical area of Ethiopia were spatially significant predictors.

## Conclusion

Overall, the consumption of foods rich in vitamin A was low and spatially non-random in Ethiopia. Poor wealth status of the household, rural residence and living tropical area were spatially significant predictors for the consumption of foods rich in vitamin A in Ethiopia. Policymakers and health planners should intervene in nutrition intervention at the identified hot spot areas to reduce the poor consumption of foods rich in vitamin A among children aged 6–23 months.

## Introduction

Vitamin A is a fat-soluble vitamin used for rhodopsin formation, a photoreceptor pigment of the retina, which helps maintain epithelial tissues, and immune enhancers [1]. Mainly performed retinol and provitamin carotenoid foods rich in vitamin A were available. Preformed retinol foods rich in vitamin A were exclusively found in animal products, and provitamin A carotenoids were found in green leafy vegetables [2]. Vitamin A deficiency (VAD) is a major public health problem in poor societies, especially in low-income countries. Vitamin A deficiency remains prevalent in South Asia (44%; 13%– 79%) and sub-Saharan Africa (48%; 25% – 75%) [3]. In 2013, 94 500 (54 200–146 800) diarrhea-related deaths and 11 200 (4300–20 500) measles-related deaths were attribute by vitamin A deficiency in sub-Saharan Africa and South Asia [3]. The estimated prevalence of vitamin A deficiency among children in Ethiopia ranges between 20% and 39% [1,3].

Low consumption of foods rich in vitamin A during nutritionally demanding periods in life, such as infancy and childhood, consequences vitamin A deficiency disorders [4]. Its deficiency was associated with measles, diarrhea, malaria and other infectious disease morbidity and mortality among children [3,5]. Dietary consumption of foods rich in vitamin A affected by the socio-economic and demographic status of the household [6,7], maternal knowledge and media exposure about infant and young child feeding (IYCF) and husband involvement in IYCF [8], maternal and husband education [7,9], and media exposure [9,10] were some of the factors significantly affecting dietary consumption. Vitamin A supplementation is associated with a clinically meaningful reduction in morbidity and mortality in children [5]. A meta-analysis of 17 trials showed that vitamin consumption reduces all-cause mortality and the overall risk of death by 24% among children. Another trial showed that consumption of foods rich in vitamin A and supplementation significantly reduced diarrhea-related mortality by 28% among children [11].

Despite the importance of vitamin, A-rich food consumption for children, the consumption of foods rich in vitamin A in Ethiopia remains low. Previous studies have shown that only 7% to 39% of children aged 6–23 months consume plant source foods [6,8,12], whereas only 12 to 24% of children aged 6–23 months consumed animal source vitamin A-rich foods in Ethiopia [6,12]. However, egg (11.0%) and meat (2.6%) were less frequently consumed foods [8]. Even

though, a paucity of information in the spatial distribution of foods rich in vitamin A, different studies evidenced that dietary diversity and malnutrition were non-random spatially in Ethiopia [13,14]

To date, different studies have been conducted in Ethiopia to assess dietary diversity among children, including foods rich in vitamin A consumption [15,16] However, there is no evidence conducted to determine the spatial distribution of dietary consumption of foods rich in vitamin A across the regions of Ethiopia. Exploring the spatial distribution of dietary consumption of foods rich in vitamin A in the regions of Ethiopia used for local specific nutrition intervention to tackle vitamin A deficiency-related child morbidity and mortality. Therefore, the objective of this study was to explore the spatial distribution of dietary consumption of foods rich in vitamin A and its spatial determinants among children aged 6–23 months in Ethiopia.

## Methods and materials

### Study design, area and period

This study is a community-based cross-sectional study conducted using the nationally representative 2016 Ethiopian Demographic and Health Survey (EDHS) dataset. Ethiopia is situated in the Horn of Africa from $3^0$ to $14^0$ and $33^0$ to $48^0$E.

### Source and study populations

The source population was all living children aged 6–23 months preceding the survey whereas, all living children aged 6–23 months living with their mother was the study population in the selected Enumeration Areas (EAs). In the 2016 EDHS, a total of 645 clusters (EAs) (202 urban and 443 rural) selected with a probability proportional to each EA size and independent selection in each sampling stratum. Among the selected clusters with zero coordinates and clusters without a proportion of children, the status of consumption of foods rich in vitamin A was excluded from the analysis. Finally, we selected a total of 598 (185 urban and 413 rural) clusters for this study. Among the selected clusters, a total of 3055 weighted number of living children aged 6–23 months living with their mother were included.

We accessed the recorded data at https://dhsprogram.com/ upon request.

### Data collection tools and procedures

Ethiopian demographic and health survey data were collected by a two-stage stratified cluster sampling technique. Each region of the country was stratified into urban and rural areas, yielding 21 sampling strata. In the first stage, 645 EAs were selected with a probability proportional to the EA size by independent selection in each sampling stratum. In the second stage, a fixed number of 28 households per cluster were selected with an equal probability of systematic sampling from the newly created household listing. The detailed sampling procedure is available in the EDHS reports from the Measure DHS website (www.dhsprogram.com).

### Outcome variable

Children were aged 6–23 months living with their mother who consumed foods rich in vitamin A (Egg, Meat, Vegetables, Green leafy vegetables, Fruits, Organ meat, and Fish) at least one food item among the seven food items at any time in the last 24 hours preceding the interview was declared good consumption of foods rich in vitamin A, whereas no consumption in the seven food items rich in vitamin A in 24 hours preceding the interview was poor consumption [17].

### Independent variables

From the 2016 EDHS datasets, the proportion of mothers' age, educational status of mother and husband, residence, the religion of mother, parity, wealth index status, media exposure, and altitude were considered independent variables.

### Operational definition

**Media exposure.**    If the respondents have a chance to listen to either radio or television declare as having media exposure if not both not have media exposure.

**Altitude.**    It is declared as, Tropical zone (Kolla)—is below 1830 meters in elevation, Subtropical (Woina Dega)—includes the highlands areas of 1830–2440 meters elevation, and Cool zone (Dega) is above 2440 meters in elevation.

### Data management and analysis

The data were cleaned by STATA version 14.1 software and Microsoft Excel. Sample weighting was performed for further analysis.

**Spatial autocorrelation and hot spot analysis.**    Spatial autocorrelation (Global Moran's I) statistic was conducted to assess whether the consumption of foods rich in vitamin A among children aged 6–23 months was dispersed, clustered, or randomly distributed in Ethiopia. Moran's I values close to −1 indicate poor consumption of foods rich in vitamin A dispersed, close to +1 indicates clustered, and if Moran's I value zero indicates randomly distributed [18]. A statistically significant Moran's I value ($p < 0.05$) had a chance to reject the null hypothesis, which indicates the presence of spatial autocorrelation. Hot spot analysis (the Getis-Ord Gi* statistic) of the z-scores and significant p-values tells the features with either hot spot or cold spot values for the clusters spatially.

### Empirical Bayesian Kriging spatial interpolation

The spatial interpolation technique is used to predict poor consumption of foods rich in vitamin A among children aged 6–23 months for unsampled areas in the country based on sampled EAs. For the prediction of unsampled EAs, we used deterministic and geostatistical empirical Bayesian Kriging spatial interpolation techniques. Empirical Bayesian Kriging relaxes the assumption of the Gaussian distribution of the observed semivariogram in the input data, which rarely holds true in practice. Empirical Bayesian Kriging interpolation works by generating a new simulated semivariogram at each location from the estimated semivariogram from the input data. The weight of the new simulated semivariogram is calculated by Bayes' rule [19].

### Spatial scan statistics

We employed Bernoulli-based model spatial scan statistics to determine the geographical locations of statistically significant clusters for poor consumption of foods rich in vitamin A among children aged 6–23 months using Kuldorff's SaTScan version 9.6 software [20]. The scanning window that moves across the study area, in which children aged 6–23 months with poor consumption of foods rich in vitamin A were taken as cases and those with good consumption were taken as controls to fit the Bernoulli model. The default maximum spatial cluster size of < 50% of the population was used as an upper limit, allowing both small and large clusters to be detected, and ignored clusters that contained more than the maximum limit with the circular shape of the window. Most likely clusters were identified using p-values and likelihood ratio tests based on the 999 Monte Carlo replications.

## Geographically weighted regression analysis

The ordinary least squares regression (OLS) model is a global model that estimates only one single coefficient per explanatory variable over the entire study area. Global models assume factors that affect poor consumption of foods rich in vitamin A were stationary geographically. The assumption of geographical independence may bias the parameter estimates. The assumption of geographical independence relaxes by geographically weighted regression analysis. A geographically weighted regression model is an extension of the OLS regression model and gives local parameter estimates to reflect changes over space in the association between an outcome and explanatory variables [18].

For the interest of geographically weighted regression analysis, the aggregated proportion of poor consumption of foods rich in vitamin A among children aged 6–23 months and all the predictor variables were calculated for each cluster. To determine the predictor variables for poor consumption of foods rich in vitamin A among children aged 6–23 months, we used a geographically weighted regression (GWR) model.

To check the assumption of spatial dependency, an explanatory analysis was performed first by Arc GIS 10.7 software. Statistically significant ($P < 0.01$) Koenker (BP) statistics indicate that the relationships are not consistent (either due to non-stationarity or heteroscedasticity). Multicollinearity (variance inflation factor $<7.5$) was checked to exclude redundancy among explanatory variables. In the case of spatial dependency, the coefficient of the independent variable varies locally, and the predictor variables may or may not be significant locally. The model structure of geographically weighted regression is written as

$$Y_i = \beta_0(u_i, \, v_i) \, + \, \Sigma_k \, \beta_k(u_i, \, v_i) \, X_{ik} \, + \, \varepsilon_I$$

Where $Y_i$ is the response variable, $(u_i, v_i)$ denotes the coordinates of the $i^{th}$ point in space, $\beta_0$ is the intercept at the $(u_i, v_i)$ coordinate, $\beta_k$ is the coefficient of the covariate X at the $(u_i, v_i)$ coordinate, and $\varepsilon_i$ is the random error term.

## Model calibration

We used Multiscale Geographically Weighted Regression (MGWR) version 2.0 software to calibrate the parameter estimates of the Geographically Weighted Regression (GWR) model [21]. The new version of GWR is termed multiscale geographically weighted regression (MGWR) and potentially provides a more flexible and scalable framework in which to examine multiscale processes. Adaptive bi-square kernels were used for geographical weighting to estimate local parameter estimates. The golden section search method was used to determine the best bandwidth size based on corrected Akaike's Information Criterion (AICc), and the bandwidth with the lowest AICc was used to determine the best fit model for local parameter estimates.

Geographical variability for each coefficient can be assessed by comparing the AICc between the GWR model and the global OLS regression model. The corrected Akaike's Information Criterion (AICc) was obtained by minimizing the Akaike Information Criteria (AIC), which is [18]:

$$AICc = 2n\log_e(\hat{\sigma}) + n\log_e(2\pi) + \{\frac{(n + \text{tr}(s))}{(n - 2 - \text{tr}(s))}\}$$

Where n is the sample size, $\hat{\sigma}$ is the estimated standard deviation of the error term, and tr(**S**) denotes the trace of the hat matrix, which is a function of the bandwidth. Finally, local parameter estimates were plotted on Arc GIS 10.7 (ESRI Inc., Redlands, CA, USA, version 10.7) software.

### Ethical consideration

We submitted a concept paper to DHS Program/ICF International Inc., and a letter of permission was confirmed from the International Review Board of Demographic and Health Surveys (DHS) program data archivist to download the dataset for this study.

## Results

### Characteristics of the respondents and study children

A total of 3055 children aged 6–23 months were included in this study. More than half (53%) of the children were females. Of the total children, 18.45% were aged 6–8 months, and 36.6% were aged 12–17 months. The mean ± SD age of the children was 13.92 ± 5.05 months. The majority (67.50%) of the mothers were in the age group of 20–34 years. The mean ± SD age of the mothers was 28.25 ± 6.47 years. Most (94%) mothers were married. Forty-four per cent of the households had poor household wealth status (Table 1).

### Vitamin A rich food consumption among children aged 6–23 months

Overall, two-thirds (62%: 95% CI: 60.56–64.00) of children aged 6–23 months had poor consumption of foods rich in vitamin A. Animal source foods were the least consumed foods in the last 24 hours in the survey period. Egg consumption was reported to the most taken food in the last 24-hours period (Table 2).

### Spatial distribution of poor consumption of foods rich in vitamin A among children aged 6–23 months

To determine spatial clustering of poor consumption of foods rich in vitamin A, global spatial statistics were estimated using Moran's I value. As shown in the figure below, statistically significant z-scores indicate at 152 km distances where spatial processes promoting clustering are most pronounced. The incremental spatial autocorrelation indicates that a total of 8 distance bands were detected with a beginning distance of 120 000 meters. The spatial distribution of poor consumption of foods rich in vitamin A among children aged 6–23 months in Ethiopia was found to be nonrandom, with a global Moran's I of 0.21 and a p-value of 0.001. For the z-score of 14.13, there is less than 1% likelihood that this high-clustered pattern could be the result of random chance (Fig 1).

### Hot spot (Getis-Ord Gi*) analysis

As shown in Fig 2 below, the red color indicates the more intense clustering of a high (hot spot) proportion with poor consumption of foods rich in vitamin A preceding the survey period. A high proportion of poor consumption of foods rich in vitamin A among children aged 6–23 months clustered in Afar, eastern Tigray, southeast Amhara, and the eastern Somali region of Ethiopia. However, Addis Ababa, Gamebela, and Central Oromia regions of Ethiopia were less risk areas for poor consumption of foods rich in vitamin A among children aged 6–23 months.

### Spatial scan statistics analysis

In spatial scan analysis, a total of 187 significant clusters were identified. As shown in Fig 3 below, the red window indicates the significant clusters. Among the significant clusters, 142 clusters most likely (primary), and 45 clusters were secondary. The most likely (primary) clusters were located at 11.626646 N, 39.666950 E in a 278.08 km radius in Afar, the eastern part of Tigray, and most of Amhara National Regional State of Ethiopia. The secondary significant

**Table 1. Sociodemographic characteristics of the respondents and study children aged 6–23 months in Ethiopia, EDHS 2016 (n = 3055).**

|  | Variables | Frequency (n) | Percent (%) |
|---|---|---|---|
| **Child age (months)** | 6–8 months | 564 | 18.45 |
|  | 9–11 months | 502 | 16.43 |
|  | 12–17 months | 1,118 | 36.59 |
|  | 18–23 months | 871 | 28.53 |
| **Child sex** | Male | 1,433 | 46.90 |
|  | Female | 1,622 | 53.10 |
| **Mother's age (years)** | <20 | 369 | 12.08 |
|  | 20–34 | 2,062 | 67.50 |
|  | 35–49 | 624 | 20.42 |
| **Marital status** | Married | 2864 | 93.76 |
|  | Not married | 191 | 6.24 |
| **Religion** | Orthodox | 1,049 | 34.35 |
|  | Muslim | 1,236 | 40.45 |
|  | Others* | 770 | 25.20 |
| **Mother's education** | No education | 1,864 | 61.03 |
|  | Primary education | 936 | 30.62 |
|  | Secondary and above | 255 | 8.35 |
| **Husband's education** | No education | 1,291 | 44.55 |
|  | Primary education | 1,198 | 41.34 |
|  | Secondary and above | 408 | 14.10 |
| **Mother occupation** | Working | 1,265 | 41.41 |
|  | Not working | 1,790 | 58.59 |
| **Husband occupation** | Working | 2,717 | 88.93 |
|  | Not working | 338 | 11.07 |
| **Family size** | Less than three | 1,650 | 54.00 |
|  | Greater than four | 1,405 | 46.00 |
| **Media exposure** | No media exposure | 2,045 | 66.92 |
|  | Had media exposure | 1,010 | 33.08 |
| **Household wealth** | Poor | 1,350 | 44.18 |
|  | Middle | 683 | 22.37 |
|  | Rich | 1,022 | 33.44 |
| **Residence** | Rural | 2,684 | 87.85 |
|  | Urban | 371 | 12.15 |
| **Altitude** | Tropical zone (Kolla) | 1,362 | 44.58 |
|  | Sub-tropical zone (Woina Dega) | 1,159 | 37.92 |
|  | Cool zone (Dega) | 534 | 17.50 |
|  | **Total** | **3055** | **100** |

* = Catholic, Protestant, Traditional.

clusters were located at 6.745502 N, 44.259010 E in a 360.64 km radius in Somali and some part of the Oromia National Regional State of Ethiopia. Children were aged 6–23 months living in the primary cluster were 46% more likely vulnerable to poor consumption of foods rich in vitamin A than outside the window (RR = 1.46, LLR = 83.78, P-value < 0.001). Children living in the secondary cluster were 36% more likely to risk poor consumption of foods rich in vitamin A than those living outside the window (RR = 1.36, LLR = 27.18, P-value < 0.001) (Table 3).

**Table 2. Consumption of foods rich in vitamin A among children aged 6–23 months in the last 24 hours preceding survey EDHS, 2016, Ethiopia (n = 3055).**

| S. No. | Food groups interviewed in the last 24 hours. | Consumption status | |
|---|---|---|---|
| | | Good (%) | Poor (%) |
| 1 | Have the child took eggs in the last 24 hours? | 16.81 | 83.19 |
| 2 | Has the child taken meat (beef, pork, lamb, chicken, etc.) in the last 24 hours? | 5.95 | 94.05 |
| 3 | Has the child taken a pumpkin, carrots, squash (yellow or orange inside) in the last 24 hours? | 12.17 | 87.83 |
| 4 | Has the child taken any dark green leafy vegetables in the last 24 hours? | 13.56 | 86.44 |
| 5 | Has the child taken mangoes, papayas, other vitamin A fruits in the last 24 hours? | 12.83 | 87.17 |
| 6 | Has the child taken liver, heart, other organs in the last 24 hours? | 3.89 | 96.11 |
| 7 | Has the child taken fish or shellfish in the last 24 hours? | 1.31 | 98.69 |
| | **Overall consumption of foods rich in vitamin A among children age 6–23 months.** | **32.70** | **62.30** |

## Prevalence of poor consumption of foods rich in vitamin A among children aged 6–23 months in Ethiopia

In most parts of Ethiopia, children aged 6–23 months, were vulnerable to poor consumption of foods rich in vitamin A. Children living in Afar, eastern Amhara, and eastern Somalia regions of Ethiopia were more vulnerable to poor consumption of foods rich in vitamin A as compared to other regions of Ethiopia (Fig 4).

## Geographically weighted regression and ordinary least squares model comparison

Selected predictor variables fitted in the geographically weighted regression model. For model compression, both the ordinary least squares (OLS) model and the geographical weighted

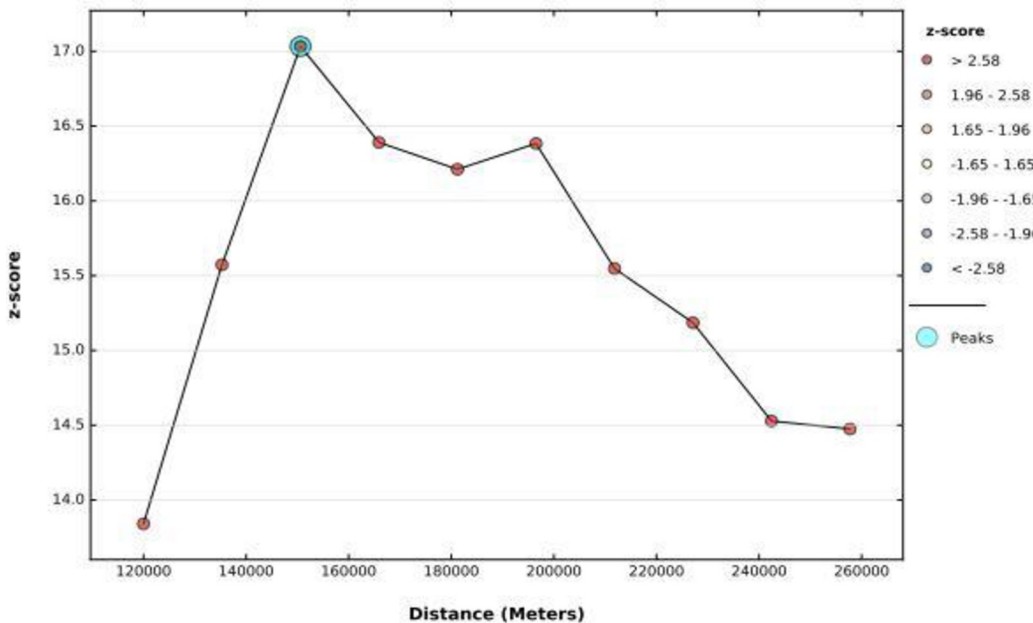

**Fig 1. The spatial autocorrelation of poor consumption of foods rich in vitamin A among children aged 6–23 months in Ethiopia.**

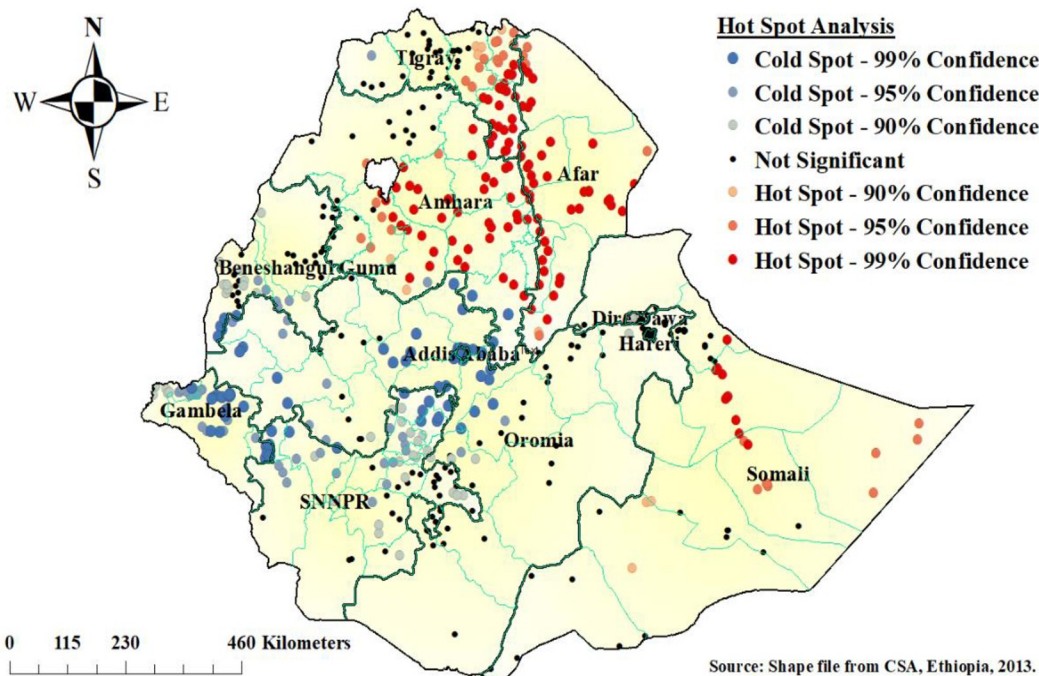

**Fig 2. Hot spot analysis of poor consumption of foods rich in vitamin A among children aged 6–23 months, in Ethiopia.**

regression (GWR) model were fitted. The bandwidth corrected Akakian Information Criteria (AICc), adjusted $R^2$, and log-likelihood were considered for model comparison. Comparing

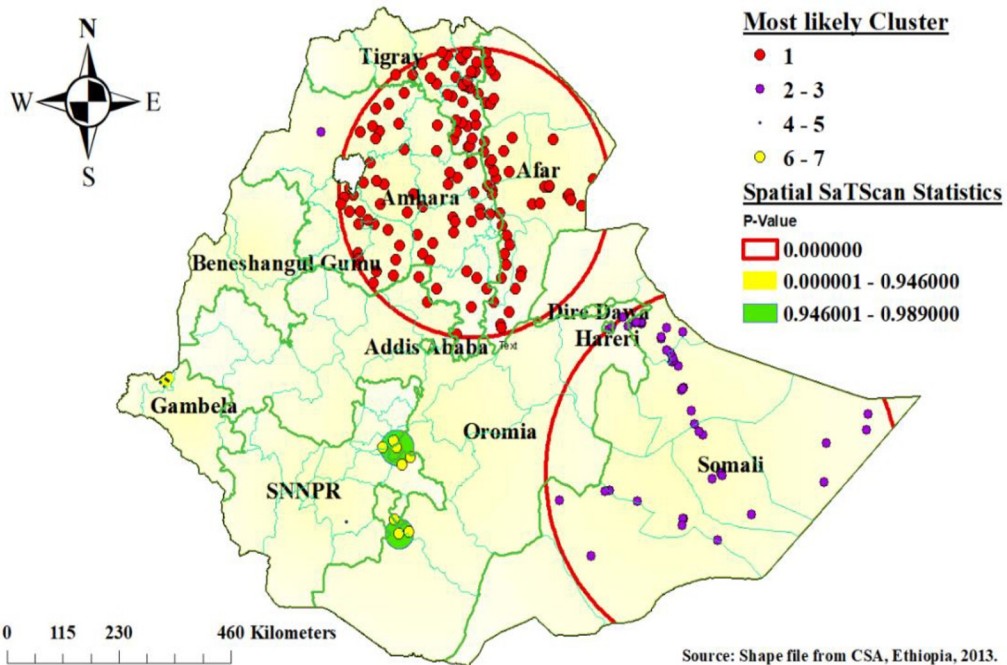

**Fig 3. Most likely (primary) and secondary cluster for poor consumption of foods rich in vitamin A among children aged 6–23 months, Ethiopia.**

**Table 3. Significant spatial scan statistics clusters of poor consumption of foods rich in vitamin A among children aged 6–23 months, EDHS, 2016.**

| Cluster type | Significant Enumeration Areas(clusters) detected | Coordinates/Radius | Populations | Cases | RR | LLR | P-value |
|---|---|---|---|---|---|---|---|
| Primary | 496, 189, 611, 571, 191, 345, 478, 254, 389, 591, 18, 200, 241, 455, 401, 354, 332, 368, 616, 344, 249, 348, 55, 617, 97, 351, 488, 544, 442, 547, 545, 300, 66, 136, 570, 627, 276, 449, 460, 128, 599, 620, 143, 334, 176, 38, 392, 205, 79, 542, 310, 283, 499, 178, 10, 267, 199, 637, 511, 102, 130, 160, 37, 206, 132, 172, 295, 135, 120, 421, 424, 427, 440, 456, 538, 632, 510, 596, 384, 572, 512, 336, 628, 605, 482, 550, 237, 94, 24, 484, 220, 201, 163, 575, 75, 403, 430, 152, 327, 158, 585, 350, 423, 99, 298, 623, 167, 579, 235, 425, 564, 429, 312, 80, 4, 127, 73, 531, 196, 355, 362, 169, 129, 382, 322, 551, 230, 640, 226, 375, 431, 51, 604, 474, 263, 156, 341, 218, 516, 121, 481, 188 | 11.627 N, 39.667 E / 278.08 km | 663 | 554 | 1.46 | 83.78 | < 0.001 |
| Secondary | 490, 543, 92, 492, 171, 198, 146, 95, 85, 358, 164, 138, 497, 521, 588, 458, 553, 278, 77, 629, 214, 318, 251, 573, 187, 239, 116, 22 33, 568, 277, 527, 269, 556, 630, 64, 439, 57, 480, 8, 210, 186, 454, 436, 566 | 6.745 N, 44.259 E / 360.64 km | 250 | 209 | 1.36 | 27.18 | < 0.001 |

NB: RR = Relative Risk, LLR = Log-Likelihood Ratio.

the global model, geographical weighted regression was the best fit model with an AICc of 1570 compared with 1616. Additionally, the GWR model best explained by the predictor variables for poor consumption of foods rich in vitamin A among children aged 6–23 months, with an adjusted $R^2$ value of 62% compared to 28% (Table 4).

## Spatial factors associated with poor consumption of foods rich in vitamin A

In the geographically weighted regression model independent variables, poor wealth status of the household, rural residence, and living in the tropical area were spatially statistically

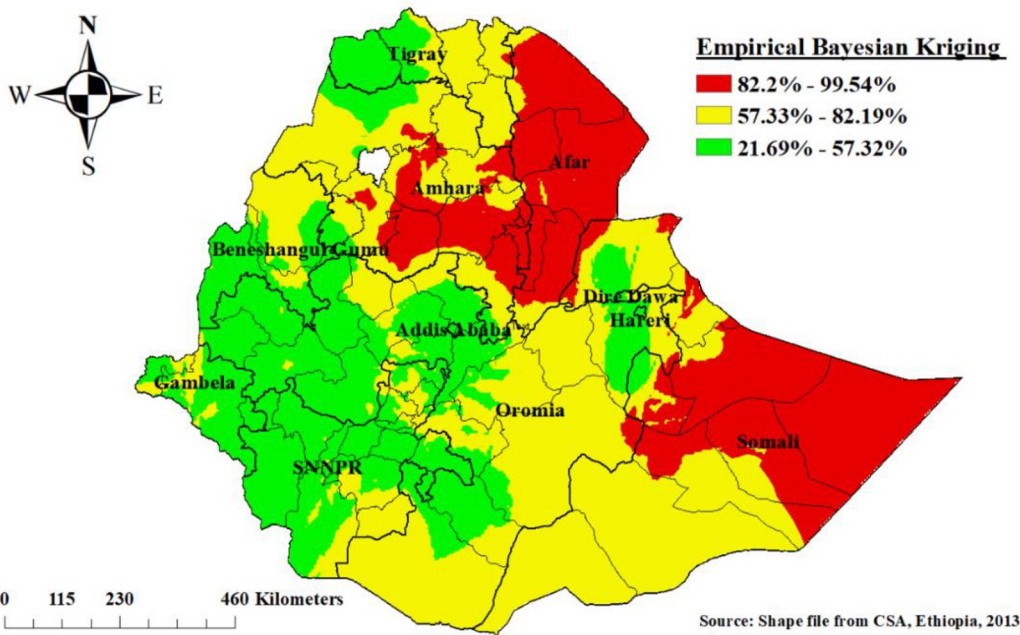

**Fig 4. Empirical Bayesian Kriging interpolation of poor consumption of foods rich in vitamin A among children aged 6–23 months, Ethiopia.**

**Table 4. Model comparison between the OLS and the GWR model.**

| Values | OLS Model | GWR Model |
|---|---|---|
| AICc | 1616.27 | 1570.65 |
| Adjusted $R^2$ | 28% | 62% |
| Log likelihood | -788.48 | -742.23 |

NB: AICc = corrected Akakian Information Criteria.

significant factors for poor consumption of foods rich in vitamin A among children aged 6 to 23 months. The strength of the association with independent variables varies spatially, and the effects of variables had a positive and negative effect spatially.

The poor wealth status of the household had different statistical significance in different parts of Ethiopia for poor consumption of foods rich in vitamin A among children aged 6 to 23 months. The coefficients of poor wealth status vary spatially between 0.088 and 0.203, indicating that the effect of association differs across regions of Ethiopia. In the significant parts of Ethiopia, a 1% increase in the poor wealth status proportion of the household increases the prevalence of poor consumption of foods rich in vitamin A among children aged 6 to 23 months by a range of 14.6% to 20.3%. Poor wealth status was not statistically significant in the Benishangul Gumez Regional state, most of the Tigray and Amhara regional states, and some of the Oromia and Gambela regional states of Ethiopia (Fig 5).

The residence was statistically significant for poor consumption of foods rich in vitamin A across regions of Ethiopia. The effect size of rural residence varies spatially from -0.072 to 0.309, which indicates that rural residents had a negative and positive effect spatially for poor consumption of foods rich in vitamin A among children aged 6–23 months. Keeping other factors constant, living in rural areas had increased the risk of poor consumption of foods rich in vitamin A by a range of 22.5% to 31% (Fig 6).

Furthermore, children living in the tropical areas of Ethiopia had different spatial significance for the poor consumption of foods rich in vitamin A. The effects of living in tropical

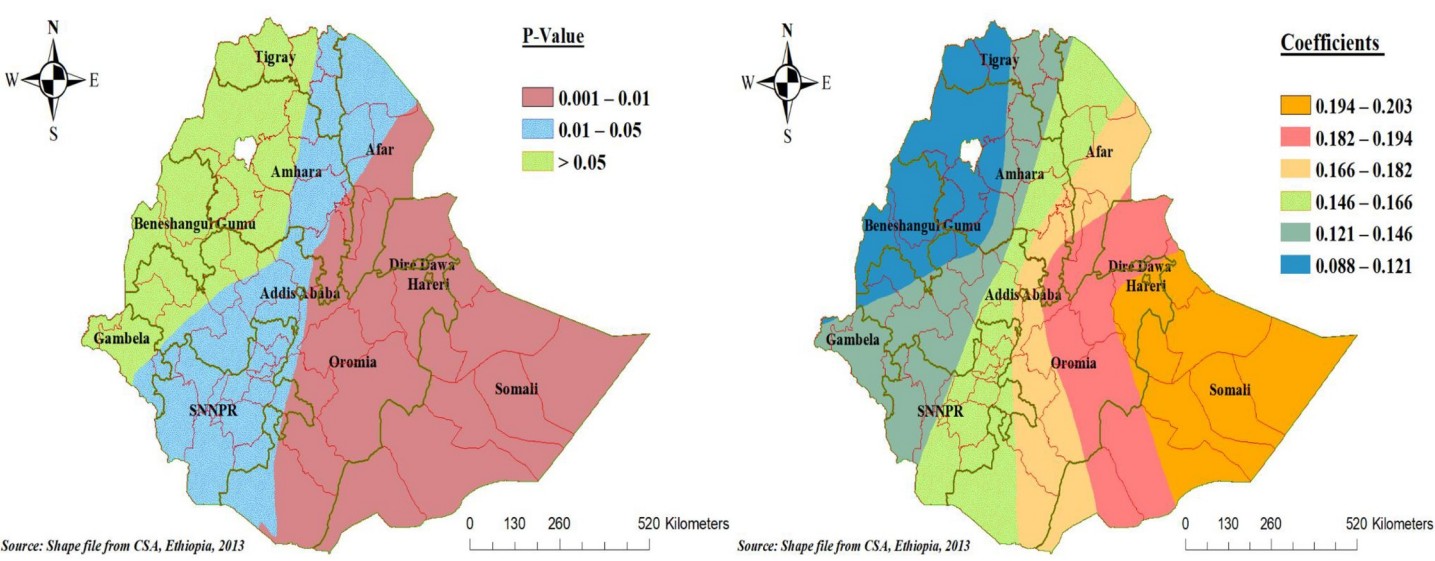

**Fig 5. Geographically varying values of significance level and coefficients per cluster for independent variable poor wealth household status of poor consumption food rich in vitamin A in the final GWR model.**

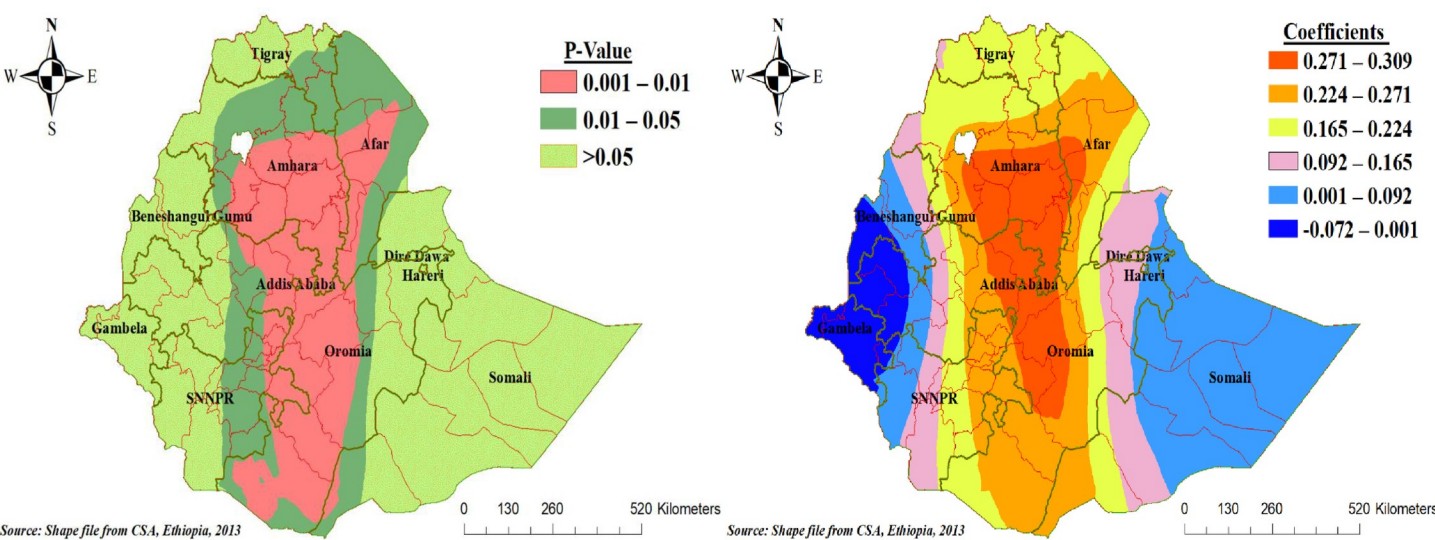

**Fig 6. Geographically varying values of significance and coefficients per cluster for the independent variable rural residence of poor consumption foods rich in vitamin A in the final GWR model.**

areas on poor consumption of foods rich in vitamin A among children aged 6–23 months vary by a range of -0.284 to 0.235, which indicates both negative and positive effects on vitamin A-rich food consumption. Keeping other factors constant, children living in the tropical area of Benishangul Gumez, the western part of Amhara and Oromia, and Gambela regions of Ethiopia had a decreased risk of poor consumption of foods rich in vitamin A by a factor of 19% to 29%, whereas children living in the tropical area of Somalia regional state of Ethiopia had an increased risk of poor consumption of foods rich in vitamin A (Fig 7).

## Discussion

This study revealed that 62.30% (95% CI: 60.56%, 64.00%) of children aged 6–23 months had poor consumption of foods rich in vitamin A in Ethiopia. The findings of this study were

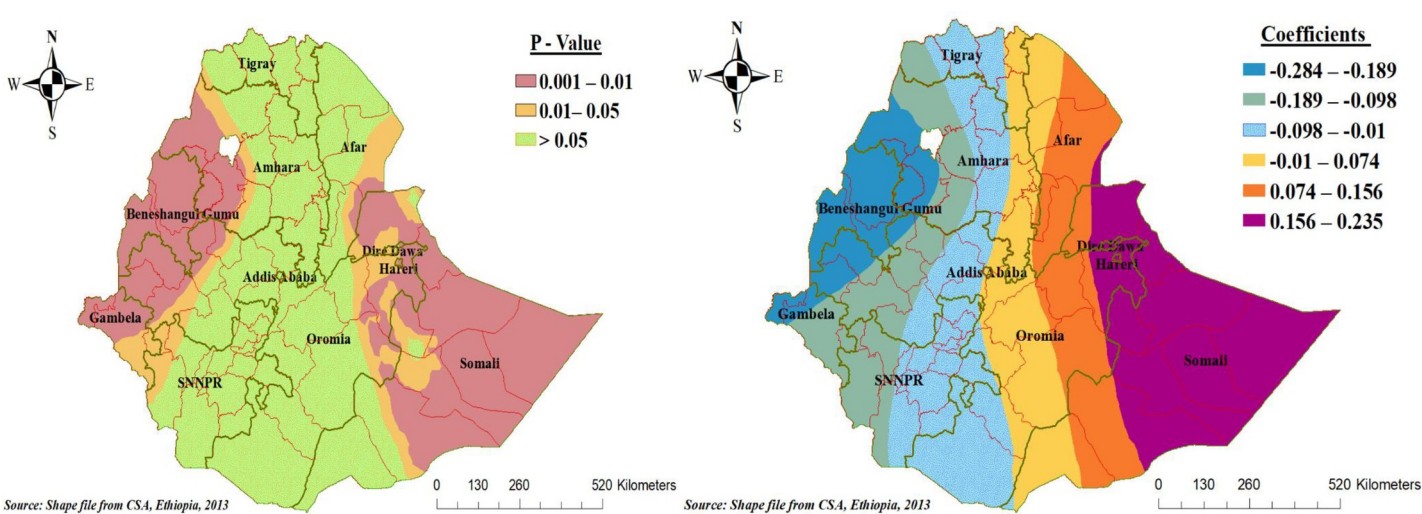

**Fig 7. Geographically varying values of significance and coefficients per cluster for the independent variable living tropical area of poor consumption foods rich in vitamin A in the final GWR model.**

lower than those of a study conducted in southern Ethiopia (71%) and the 2011 Ethiopia Demographic Health survey report (74%) [6,12]. This finding was higher than that of the study conducted in Gorche district Ethiopia, the 2014 Demographic Health survey report of Kenya (28%) and Ghana (33%), and the 2016 Demographic Health survey report of India (56%) [8,22–24].

Our study showed that the spatial distribution of poor consumption of foods rich in vitamin A was non-random in Ethiopia. Poor consumption of foods rich in vitamin A highly clustered in Afar, eastern Tigray, southeast Amhara, and the eastern Somali Regional State of Ethiopia. In line with this high proportion of clustering, spatial scan statistics analysis revealed that 187 significant clusters were identified. A high proportion of the poor consumption of foods rich in vitamin A in the Tigray and Amhara Regional states was supported by the Ethiopia National Food Consumption Survey report [25]. Even though there is no spatial analysis conducted among children aged 6–23 months, it is fair to interpret spatial studies conducted in another nutrition status. The spatial distribution of vitamin A-rich food consumption has supported a study conducted in Ethiopia which is childhood undernutrition was clustered at Northern, Middle, North East and North West areas of Ethiopia particularly from all administrative zones of Amhara, Tigray, and Afar [13]. Besides, in a study conducted in South Africa, the spatial distribution of nutritional status among childhood period was non-random geographically [26]. Furthermore, a recent study conducted on the spatial distribution of iron-rich food consumption in Ethiopia showed that spatial clustering was consistent with this finding [27]. The observed geographical variation of poor consumption of foods rich in vitamin A across regions of Ethiopia might be due to the regional variation in dietary preference, low practice to complementary feeding, socioeconomic status, demographic factors such as pastoralist region, and seasonal differences for the consumption of fruits and vegetables [28].

The local parameter estimates of the predictor variables of the model fit vary spatially in Ethiopia. Poor wealth status of the household, rural residence, and living in the tropical area of Ethiopia were statistically significant local independent variables for poor consumption of foods rich in vitamin A among children aged 6–23 months.

Our study revealed that the poor household wealth status was a spatially statistically significant predictor variable for poor consumption of foods rich in vitamin A. In the significant parts of Ethiopia, a 1% increase proportion of poor wealth status of the household could increase the prevalence of poor consumption of foods rich in vitamin A among children aged 6–23 months 14.6% to 20.3%. The findings of this study are supported by previous studies in Ethiopia [7,10,15], which evidenced that children born from the richest household had adequate dietary diversity and male frequency. Another study in Nepal [29], showed that children from the poorest household wealth quintile had higher odds of not consuming legumes and nuts, dairy products, flesh foods, other fruits and vegetables and did not meet the minimum dietary diversity. This finding is not similar to a study done in Ethiopia, which found that the household wealth status is not statistically significant for the consumption of foods rich in vitamin A [6]. The possible reason for its inconstancy might be setting, time, and sample size deference. The possible reason might be that households with poor wealth did not obtain minimum meal frequency for their child, and poor household wealth will affect adherence to the consumption of foods rich in vitamin A and dietary diversity to their child [30].

Another factor spatially affecting consumption of foods rich in vitamin A was residence. Living in rural areas increased the risk of poor consumption of foods rich in vitamin A by a range of 22.5% to 31%. This study is consistent with the study done in Ethiopia, which found that children living in rural areas had poor consumption of dietary diversity [31]. The possible justification might be living in rural areas had no access to foods rich in vitamin A, poor knowledge about foods rich in vitamin A, and other socioeconomic factors [32].

Furthermore, this study revealed that children living in the tropical area of Ethiopia were spatially significant for poor consumption of foods rich in vitamin A. Children living in the tropical area of Benishangul, the western part of Amhara, Oromia, and Gambela region of Ethiopia had less risk of poor consumption of foods rich in vitamin A by a range of 19% to 29%, whereas children living in the tropical area of Somalia regional state of Ethiopia had poor consumption of vitamin A-rich foods. The possible discrepancy might be the different accessibility of fruit and vegetable vitamin A rich foods and different cultural and behavioral practices in the feeding of the child in these different regions of Ethiopia.

### Strength and limitation of the study

As Tobler's first law of geography states, "Everything is related to everything else, but near things are more related than distant things"[33]. Based on Tobler's first law of geography, poor consumption of foods rich in vitamin A was spatially autocorrelated. In the presence of spatial dependence and heterogeneity, the estimates obtained from the global model would be biased. Therefore, fitting the GWR model and knowing the spatial distribution of poor consumption of foods rich in vitamin A in the regions of Ethiopia provides important insight to policymakers and health planners and valuable hot spot maps used for more effective and cost-efficient nutrition intervention.

The limitation of this study was that the coordinates collected at the cluster level were not individual-level which is difficult to do at the individual level, and clusters without coordinates that were deleted may not be representative. As well, the consumption status of vitamin A-rich foods were measured 24 hours recall will lead to recall and social desirability bias.

## Conclusion and recommendations

In Ethiopia, poor consumption of foods rich in vitamin A varies geographically across the regions of Ethiopia. Spatially statistically significant hot spots of poor consumption of foods rich in vitamin A were identified in Afar, eastern Tigray, southeast Amhara, and the eastern Somali region of Ethiopia, whereas Addis Ababa, Gamebela, and Central Oromia regions of Ethiopia were less risk areas. This study showed that predictor variables for poor consumption of foods rich in vitamin A vary spatially in Ethiopia. Poor wealth status of the household, rural residence and living tropical area were spatially statistically significant predictors across different regions of Ethiopia. Therefore, policymakers and health planners should design nutrition intervention programs at the identified hot spot areas to reduce the poor consumption of foods rich in vitamin A among children.

### The implications for policymakers and researchers

The results of this study provide a rich understanding of the spatial distribution consumption of foods rich in vitamin A among children aged 6–23 months in Ethiopia. The finding of hot spot maps in line with scan statistics analysis across Ethiopia used to policymakers to give direct nutrition intervention locally. This study focuses on the typical consumption of foods rich in vitamin A. The association between dependent and independent variables might vary across different parts geographically. In such situations, fitting the global model may bias the parameter estimates. With this concept, other researchers should apply the GWR model to assess other nutritional problems and dietary diversity among children and reproductive age group mothers, particularly micronutrient insecure areas.

## Supporting information

**S1 File.**
(RAR)

## Acknowledgments

We, Authors, acknowledged the Demographic and Health Surveys (DHS) program for the accusation of the dataset.

## Author Contributions

**Conceptualization:** Sofonyas Abebaw Tiruneh, Dawit Tefera Fentie, Seblewongel Tigabu Yigizaw, Asnakew Asmamaw Abebe, Kassahun Alemu Gelaye.

**Data curation:** Sofonyas Abebaw Tiruneh, Dawit Tefera Fentie, Seblewongel Tigabu Yigizaw, Asnakew Asmamaw Abebe.

**Formal analysis:** Sofonyas Abebaw Tiruneh, Dawit Tefera Fentie, Seblewongel Tigabu Yigizaw, Asnakew Asmamaw Abebe, Kassahun Alemu Gelaye.

**Investigation:** Sofonyas Abebaw Tiruneh, Dawit Tefera Fentie, Kassahun Alemu Gelaye.

**Methodology:** Sofonyas Abebaw Tiruneh, Dawit Tefera Fentie, Seblewongel Tigabu Yigizaw, Asnakew Asmamaw Abebe, Kassahun Alemu Gelaye.

**Project administration:** Kassahun Alemu Gelaye.

**Resources:** Sofonyas Abebaw Tiruneh.

**Supervision:** Kassahun Alemu Gelaye.

**Validation:** Sofonyas Abebaw Tiruneh, Dawit Tefera Fentie, Kassahun Alemu Gelaye.

**Visualization:** Sofonyas Abebaw Tiruneh, Dawit Tefera Fentie, Seblewongel Tigabu Yigizaw, Asnakew Asmamaw Abebe, Kassahun Alemu Gelaye.

**Writing – original draft:** Sofonyas Abebaw Tiruneh, Dawit Tefera Fentie, Seblewongel Tigabu Yigizaw, Asnakew Asmamaw Abebe.

**Writing – review & editing:** Sofonyas Abebaw Tiruneh, Dawit Tefera Fentie, Seblewongel Tigabu Yigizaw, Asnakew Asmamaw Abebe, Kassahun Alemu Gelaye.

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
