## [Decision Letter · Decision Letter 0]

26 Nov 2020

PONE-D-20-16369

Spatial distribution and geographical heterogeneity factors associated with poor consumption of foods rich in vitamin A among children age 6 -23 months in Ethiopia: Geographical weighted regression analysis

PLOS ONE

Dear Dr. Tiruneh,

Thank you for submitting your manuscript to PLOS ONE. After careful consideration, we feel that it has merit but does not fully meet PLOS ONE’s publication criteria as it currently stands. Therefore, we invite you to submit a revised version of the manuscript that addresses the points raised during the review process.

I would like to sincerely apologise for the delay you have incurred with your submission. It has been exceptionally difficult to secure reviewers to evaluate your study. We have now received two completed reviews; their comments are available below.

Please revise the manuscript to address all the reviewer's comments in a point-by-point response in order to ensure it is meeting the journal's publication criteria. Please note that the revised manuscript will need to undergo further review, we thus cannot at this point anticipate the outcome of the evaluation process.

We look forward to receiving your revised manuscript.

Kind regards,

Miquel Vall-llosera Camps

Senior Editor

PLOS ONE

Journal Requirements:

2.We suggest you thoroughly copyedit your manuscript for language usage, spelling, and grammar. If you do not know anyone who can help you do this, you may wish to consider employing a professional scientific editing service.  

3.Thank you for stating the following in the Acknowledgments Section of your manuscript:

[We, authors, acknowledge The Demographic and Health Surveys (DHS) Program funded by the

U.S. Agency for International Development (USAID) for the accusation dataset.]

 [The author(s) received no specific funding for this work.]

4.We note that [Figure(s) 2, 3, 4, 5, 6 and 7] in your submission contain map images which may be copyrighted. All PLOS content is published under the Creative Commons Attribution License (CC BY 4.0), which means that the manuscript, images, and Supporting Information files will be freely available online, and any third party is permitted to access, download, copy, distribute, and use these materials in any way, even commercially, with proper attribution. For these reasons, we cannot publish previously copyrighted maps or satellite images created using proprietary data, such as Google software (Google Maps, Street View, and Earth). For more information, see our copyright guidelines: http://journals.plos.org/plosone/s/licenses-and-copyright.

1.    You may seek permission from the original copyright holder of Figure(s) [2, 3, 4, 5, 6 and 7] to publish the content specifically under the CC BY 4.0 license. 

**Comments to the Author**

1. Is the manuscript technically sound, and do the data support the conclusions?

Reviewer #1: Yes

Reviewer #2: Partly

2. Has the statistical analysis been performed appropriately and rigorously? 

Reviewer #1: Yes

Reviewer #2: No

3. Have the authors made all data underlying the findings in their manuscript fully available?

Reviewer #1: Yes

Reviewer #2: Yes

4. Is the manuscript presented in an intelligible fashion and written in standard English?

Reviewer #1: No

Reviewer #2: No

5. Review Comments to the Author

Reviewer #1: It is an interesting article whose findings can help policymakers and health planners to select appropriate interventions. The methodology used can be also used to identify hot spot areas of low consumption of vitamin A and its determinants not only in Ethiopia but also in other Sub-Sahara Africa countries. However, there are some caveats and extensive editing of English language and style will be needed to facilitate the reading of the whole document.

Background

It is difficult to follow the different paragraph, although the background section provides sufficient information and include relevant references.

The background section needs to be reorganized. It will be helpful to combine some paragraphs. For example, the first three paragraphs can be reduced to one.

Results

Table 1: Since the study is related to children, it will be better to present children characteristics before the mothers’ results.

Discussion

It will be important not to repeat most of results in the discussion section.

The consumption status of Foods rich in Vitamin A among children age 6-23 months was determined using a 24-hour recall. The results need to be analyzed with precaution, as it does not reflect any dietary habits. Thus, it needs to be highlighted in Discussion.

It will be helpful to give some examples of regional variation dietary preference, low practice to complementary feeding, or socioeconomic status.

References for this confirmation: The possible justification might be living in rural area had no access to get foods rich in vitamin A, poor knowledge about foods rich in vitamin A, and other socioeconomic factors.

Reviewer #2: This study aimed to assess the spatial distribution and its determinants of dietary consumption of foods rich in vitamin A among children aged 6-23 months in Ethiopia. It could provide valuable information to identify areas with high vitamin A deficiency. However, there are a number of issues to be addressed.

Abstract

- Methods: please specify how poor vitamin A consumption was defined

Background

- Need references for the sentence ‘So far, different studies conducted in Ethiopia to assess dietary diversity among children including foods rich in vitamin A consumption’.

- The authors presented the importance of vitamin A in the background. However, the background regarding geospatial analysis is relatively weak. The authors should provide more details to strengthen the justification of this study. For instance, the authors indicated that there was no evidence on geospatial distribution of dietary consumption in Ethiopia. However, relevant studies were done from other countries so it would be informative to add- what is the current knowledge on geospatial distribution of dietary intake, specifically in Africa, what is the gap and how this study could contribute to the existing body of evidence. Also, if there is no study on spatial distribution of dietary consumption of foods rich in vitamin A, it would be still useful to add studies in Ethiopia targeting other nutrients or other nutrition outcomes such as stunting and wasting.

- ‘The magnitude of vitamin A deficiency (VAD) was highest in Sub-Saharan Africa (48%; 25–75) and South Asia (44%; 13–79)’: if 25-75, 13-59 mean CI or other, please specify

Methods and materials

- I suggest the authors revise ‘Source and study populations’ and ‘Data collection tools and procedures’ as some parts were overlapped and it is not easy to understand

- Some important information is missing regarding the data source- for instance, how did this study handle missing data and what was the survey response rate?

- Outcome variable: as it is critical part in the manuscript, the authors need to provide enough details such as who responded to the question, what were the seven food items and what was the justification to define poor consumption of vitamin A

- Predictor variables: It is not clear how the authors considered possible multicollinearity

- It is not clear how the authors considered complex survey design

- Data management and analysis: didn’t the author also use Kuldorff’s SaTScan version 9.6 software, Arc GIS 10.7 software and MGWR (Multi-scale Geographically Weighted Regression)? If then, please indicate in addition to STATA and Excel

- It is not clear what was done for model validity and uncertainty assessment. Please provide details.

- Ethical consideration: better to clearly say that ethical approval was not required for what reason.

Results

- Be consistent with presenting numbers- up to two decimals, one or? i.e. 18.45%, 61%

- Table 1: the authors need to explain how variables were classified in the ‘Methods’ section. For instance, how household wealth was classified into poor, middle and rich? Is it solely based on household income or with other assets? Please explain what Dega means. Also, how ‘media exposure’ was defined?

- How about the associations with other variables such as - education, religion, occupation, child age, etc. and outcome? Please also specify if the association were not significant

Discussion

- It is not clear what the first paragraph is trying to say

- The last sentence on page 19 needs more elaboration- how dietary preference, low practice to complementary feeding or socioeconomic status differ by regions and how it could explain geographical variation of vitamin A consumption. Same goes for the last sentence on page 20.

- It is not clear what ‘The possible reason might be household with poor wealth did not get minimum meal frequency to their child and poor wealth will affect adherent to the consumption of foods rich in vitamin A and dietary diversity to their child.’ means, please specify.

- The authors could have provided comprehensive comparison with other studies to strengthen the discussion part. For instance, what were the results of other similar studies examining spatial distribution of food consumption or nutrition/health status? How similar or different were the results and what would be the possible reasons for that?

- The authors listed one limitation but there might be more – for instance, how food consumption was defined as poor or good? Was amount of food considered? Was there any possibility of recall bias?

References

- Need revision. For instance, #3 ref: (World Health Orgnaisation) is repeated

- #10 ref: year is repeated twice like ‘J Nutr Metab. 2013;2013’

There are some grammar and flow issues so I recommend copyediting. Below are some examples

- Venerable: do you mean vulnerable? (appeared several times in abstract, result)

- The following sentence on p9 is not clear- On the other hand, only 12 to 24% of children age 6-23 months consumed animal source foods rich in vitamin A in Ethiopia (6,12), however, eggs (11.0%) and meat (2.6%) were less frequently consumed (8).

- p20 …male frequency and Nepal which is Children from the poorest… did you mean ‘meal’?

- p21 .. The expectation of the finding of hos spot maps in line.. did you mean ‘hot’?

- p21 …This study focuses on typical consumption of foods rich in vitamin A.Tthe association between….

6. PLOS authors have the option to publish the peer review history of their article (what does this mean?). If published, this will include your full peer review and any attached files.

Reviewer #1: No

Reviewer #2: No

---

## [Author Response · Author response to Decision Letter 0]

30 Dec 2020

Response to Reviewers’‎

Spatial distribution and geographical heterogeneity factors ‎associated with poor consumption of foods rich in vitamin A ‎among children aged 6 -23 months in Ethiopia: Geographical ‎weighted regression analysis ‎

Sofonyas Abebaw Tiruneh1*, Dawit Tefera Fentie2, Seblewongel Tigabu Yigizaw 2, Asnakew ‎Asmamaw Abebe2, Kassahun Alemu Gelaye2.‎

The authors, extending our great thanks for the editors and reviewers for this manuscript as the ‎stand of this review. The comments raised by the reviewers and editors are vital and defiantly it ‎will improve the quality of the manuscript. We have addressed all the issues raised by the ‎reviewers and editors point-by-point response and believed that the revised version of the ‎manuscript is satisfactory and will meet the journal publication requirements. As well, the journal ‎requirements amended accordingly the journal submission guideline. Please note that words and ‎sentences highlighted by Areal font under the reviewers' question and comment were the authors' ‎response and reaction for each issue. ‎

Stay Safe!!!‎

The Authors.‎

‎ 

‎ 

Review Comments to the Author

Reviewer #1: It is an interesting article whose findings can help policymakers and health planners ‎to select appropriate interventions. The methodology used can be also used to identify hot spot ‎areas of low consumption of vitamin A and its determinants not only in Ethiopia but also in other ‎Sub-Sahara Africa countries. However, there are some caveats and extensive editing of English ‎language and style will be needed to facilitate the reading of the whole document.‎

Thank you for the comment!‎

Background

It is difficult to follow the different paragraph, although the background section provides ‎sufficient information and include relevant references. The background section needs to be ‎reorganized. It will be helpful to combine some paragraphs. For example, the first three ‎paragraphs can be reduced to one.‎

Noted! Modified accordingly!‎

Results

Table 1: Since the study is related to children, it will be better to present children characteristics ‎before the mothers’ results.‎

Noted Thank you! it was corrected accordingly. ‎

Discussion

It will be important not to repeat most of results in the discussion section.‎

The consumption status of Foods rich in Vitamin A among children age 6-23 months was ‎determined using a 24-hour recall. The results need to be analyzed with precaution, as it does not ‎reflect any dietary habits. Thus, it needs to be highlighted in Discussion.‎

It will be helpful to give some examples of regional variation dietary preference, low practice to ‎complementary feeding, or socioeconomic status.‎

References for this confirmation: The possible justification might be living in rural area had no ‎access to get foods rich in vitamin A, poor knowledge about foods rich in vitamin A, and other ‎socioeconomic factors.‎

‎ Thank You! it was confirmed. ‎

Reviewer #2: This study aimed to assess the spatial distribution and its determinants of dietary ‎consumption of foods rich in vitamin A among children aged 6-23 months in Ethiopia. It could ‎provide valuable information to identify areas with high vitamin A deficiency. However, there ‎are a number of issues to be addressed.‎

Thank you for the comment!‎

Abstract

‎- Methods: please specify how poor vitamin A consumption was defined

Noted! Corrected accordingly!‎

Background

‎- Need references for the sentence ‘So far, different studies conducted in Ethiopia to assess ‎dietary diversity among children including foods rich in vitamin A consumption’.‎

Noted corrected accordingly. ‎

‎- The authors presented the importance of vitamin A in the background. However, the ‎background regarding geospatial analysis is relatively weak. The authors should provide more ‎details to strengthen the justification of this study. For instance, the authors indicated that there ‎was no evidence on geospatial distribution of dietary consumption in Ethiopia. However, ‎relevant studies were done from other countries so it would be informative to add- what is the ‎current knowledge on geospatial distribution of dietary intake, specifically in Africa, what is the ‎gap and how this study could contribute to the existing body of evidence. Also, if there is no ‎study on spatial distribution of dietary consumption of foods rich in vitamin A, it would be still ‎useful to add studies in Ethiopia targeting other nutrients or other nutrition outcomes such as ‎stunting and wasting.‎

Thank you for your insight. It was modified accordingly. Even though there is no ‎geostatistical evidence in vitamin A rich food consumption, it will be fair to discuss ‎other nutrition status studies. ‎

‎- ‘The magnitude of vitamin A deficiency (VAD) was highest in Sub-Saharan Africa (48%; 25–‎‎75) and South Asia (44%; 13–79)’: if 25-75, 13-59 mean CI or other, please specify‎

Thank you! it was compared with other countries or continents from the evidence of the ‎study. ‎

‎ ‎

Methods and materials

‎- I suggest the authors revise ‘Source and study populations’ and ‘Data collection tools and ‎procedures’ as some parts were overlapped and it is not easy to understand

It was corrected accordingly. Since the data were secondary, the method of data ‎collection copied from the source file. ‎

‎- Some important information is missing regarding the data source- for instance, how did this ‎study handle missing data and what was the survey response rate?‎

The missing data by nature system missing. After extraction from the original dataset ‎missing data was not a problem. This data set has system missing and it was done ‎complicate case analysis. Regarding the survey response rate, so far the data were ‎nationally representative survey and it was calculated proportionally for each region. ‎Therefore, the survey was conducted in 16,650 residential households, 5,232 in urban ‎areas and 11,418 in rural areas. The sample was expected to generate an estimated ‎‎16,663 completed interviews with women age 15-49, 5,514 in urban areas and 11,149 ‎in rural areas, and 14,195 completed interviews with men age 15-59, with 4,472 in ‎urban areas and 9,723 in rural areas.‎

‎- Outcome variable: as it is critical part in the manuscript, the authors need to provide enough ‎details such as who responded to the question, what were the seven food items and what was the ‎justification to define poor consumption of vitamin A

Thank you for your insight. It was corrected accordingly the comment. The operation ‎definition was clearly stated in the manuscript. The seven food items were: Egg, Meat, ‎Vegetables, Green leafy vegetables, Fruits, Organ meat, and Fish. If the mothers or ‎caregivers respond for their child at least one food ‎item among the seven food items at ‎any time in the last 24 hours preceding the interview was ‎declared good consumption ‎of foods rich in vitamin A, if not poor consumption. ‎

‎- Predictor variables: It is not clear how the Authors considered possible multicollinearity.‎

Multicollinearity was considered for each independent variable was checked using ‎ArcGIS explanatory analysis. Therefore, all independent variables multicollinearity ‎‎(Variance Inflation Factor <7.5) from the explanatory analysis. ‎

‎- It is not clear how the authors considered complex survey design

The study design is multistage. The design was not selected by the authors because it ‎is secondary data. ‎

‎- Data management and analysis: didn’t the author also use Kuldorff’s SaTScan version 9.6 ‎software, Arc GIS 10.7 software and MGWR (Multi-scale Geographically Weighted ‎Regression)? If then, please indicate in addition to STATA and Excel

The data management (data cleaning) was done using STATA software and Microsoft ‎Excel. But spatial analysis was performed using Kuldorff’s SaTScan version 9.6 ‎software, Arc GIS 10.7 software and MGWR software. ‎

‎- It is not clear what was done for model validity and uncertainty assessment. Please provide ‎details.‎

The model validity assessment was assessed using AICc for best fit model selection.‎

‎ ‎

‎- Ethical consideration: better to clearly say that ethical approval was not required for what ‎reason.‎

The ethical clearance was weived form DHS data archivist after requesting a concept ‎paper. The dataset was publically available after submitted to a concept paper. ‎

‎ ‎

Results

‎- Be consistent with presenting numbers- up to two decimals, one or? i.e. 18.45%, 61%‎

Noted. thank you! corrected accordingly! But for the case of "61% of mothers and 45% ‎of husbands, 88% of children, and 45% " since it is numbering person rounding to the ‎nearest integer is appropriate. ‎

‎- Table 1: the authors need to explain how variables were classified in the ‘Methods’ section. For ‎instance, how household wealth was classified into poor, middle and rich? Is it solely based on ‎household income or with other assets?‎

The original dataset (secondary data) classify the wealth status of the household was ‎classified as poorest, poor, middle, rich, and richest. For further analysis (Modelling) it ‎was recategorized as Poor (Poorest and poor), middle, and rich (Rich and richest).‎

‎ Please explain what Dega means.‎

The altitude was classified as into three categories. which as, Kolla (Tropical zone) - is ‎below 1830 metres in elevation, Woina dega (Subtropical zone) - includes the ‎highlands areas of 1830 - 2440 metres, and Dega (Cool zone) - is above 2440 metres ‎in elevation. According to this classification, the Continous variable categorized as ‎such. ‎

And it will include in the operational definition. ‎

‎ Also, how ‘media exposure’ was defined?‎

The operation definition for media exposure declared as if the respondent has to ‎access to listen to either radio or television said to be having media exposure. ‎

‎- How about the associations with other variables such as - education, religion, occupation, child ‎age, etc. and outcome? Please also specify if the association were not significant

‎ Noted! These variables are not significant spatially with the outcome variable at P-‎value < 0.05. though it is no need to discuss. ‎

Discussion

‎- It is not clear what the first paragraph is trying to say.‎

The first paragraph is said to be restating the prevalence of vitamin A-rich foods ‎consumption among the study groups for internal comparison. ‎

‎- The last sentence on page 19 needs more elaboration- how dietary preference, low practice to ‎complementary feeding or socioeconomic status differ by regions and how it could explain the ‎geographical variation of vitamin A consumption. Same goes for the last sentence on page 20.‎

Noted. Thank you. This sentence elaborates the situational feeding practice of the ‎Ethiopian population. Ethiopia is a multi diversity and multiethnicity country which has ‎different regions and nation and nationalities across each region. Besides, Ethiopia ‎has four agrarian regions and five pastoralist region. Therefore the way to accessibility, ‎cultural practice on their feeding will differ across this situation. Then, this might be a ‎possible justification for the geographical variation of feeding practice among their ‎child. ‎

‎- It is not clear what ‘The possible reason might be household with poor wealth did not get ‎minimum meal frequency to their child and poor wealth will affect adherent to the consumption ‎of foods rich in vitamin A and dietary diversity to their child.’ means, please specify.‎

Noted, As we know poor wealth affects the affordability of foods for their family. ‎Therefore, households with poor wealth status will be food insecure for their family. ‎Then, this possible explanation will relate to this scenario. And, we put this regard as a ‎possible explanation for poor Vitamin A rich food consumption about. ‎

‎ ‎

‎- The authors could have provided comprehensive comparison with other studies to strengthen ‎the discussion part. For instance, what were the results of other similar studies examining the ‎spatial distribution of food consumption or nutrition/health status? How similar or different were ‎the results and what would be the possible reasons for that?‎

Noted! In such regard, there is no sufficient study on the spatial distribution of vitamin ‎A rich food consumption. But, we try to make a comparison as maximum potential.‎

‎- The authors listed one limitation but there might be more – for instance, how food consumption ‎was defined as poor or good? Was amount of food considered? Was there any possibility of ‎recall bias?‎

Thank you, we correct accordingly. ‎

‎ ‎

References

‎- Need revision. For instance, #3 ref: (World Health Orgnaisation) is repeated

‎- #10 ref: year is repeated twice like ‘J Nutr Metab. 2013;2013’‎

Thank You! we correct accordingly. ‎

There are some grammar and flow issues so I recommend copyediting. Below are some examples

‎- Venerable: do you mean vulnerable? (appeared several times in abstract, result)‎

‎- The following sentence on p9 is not clear- On the other hand, only 12 to 24% of children age 6-‎‎23 months consumed animal source foods rich in vitamin A in Ethiopia (6,12), however, eggs ‎‎(11.0%) and meat (2.6%) were less frequently consumed (8).‎

‎- p20 …male frequency and Nepal which is Children from the poorest… did you mean ‘meal’?‎

‎- p21 .. The expectation of the finding of hos spot maps in line.. did you mean ‘hot’?‎

‎- p21 …This study focuses on typical consumption of foods rich in vitamin A.Tthe association ‎between…. 

Noted! Thank You very much for your bird eye view review for this manuscript. ‎

Thank you for your constructive comment!!!‎

---

## [Decision Letter · Decision Letter 1]

8 Apr 2021

PONE-D-20-16369R1

Spatial distribution and geographical heterogeneity factors associated with poor consumption of foods rich in vitamin A among children age 6 -23 months in Ethiopia: Geographical weighted regression analysis

PLOS ONE

Dear Dr. Tiruneh,

Thank you for submitting your manuscript to PLOS ONE. After careful consideration, we feel that it has merit but does not fully meet PLOS ONE’s publication criteria as it currently stands. Therefore, we invite you to submit a revised version of the manuscript that addresses the points raised during the review process.

The manuscript has been greatly improved from the preceeding submission, as can be seen from the reviewer comments. However there are still some outstanding issues that need to be addressed before a final decision can be made. These are:

There are still major issues with the English language at various points in the manuscript - please make sure that there is a further round of proof-reading before resubmissionThe reviewer comments need to be addressed in fullThe first introductory paragraph (p3) highlights what Vitamin A is. The information about its uses is not referred back to in the rest of the document, so it is strongly suggested to remove this information, or not to put about rhodopsin etc in the first paragraph. The paper is about the spatial distribution and determinants of low Vitamin A, and not about the biology behind it so the introduction (and whole paper) should reflect this.

We look forward to receiving your revised manuscript.

Kind regards,

Andrew Amos Channon, PhD

Academic Editor

PLOS ONE

Journal Requirements:

Reviewers' comments:

Reviewer's Responses to Questions

**Comments to the Author**

1. If the authors have adequately addressed your comments raised in a previous round of review and you feel that this manuscript is now acceptable for publication, you may indicate that here to bypass the “Comments to the Author” section, enter your conflict of interest statement in the “Confidential to Editor” section, and submit your "Accept" recommendation.

Reviewer #2: (No Response)

Reviewer #3: All comments have been addressed

2. Is the manuscript technically sound, and do the data support the conclusions?

Reviewer #2: (No Response)

Reviewer #3: Yes

3. Has the statistical analysis been performed appropriately and rigorously? 

Reviewer #2: (No Response)

Reviewer #3: Yes

4. Have the authors made all data underlying the findings in their manuscript fully available?

Reviewer #2: (No Response)

Reviewer #3: Yes

5. Is the manuscript presented in an intelligible fashion and written in standard English?

Reviewer #2: (No Response)

Reviewer #3: Yes

6. Review Comments to the Author

Reviewer #2: Congratulations on this work. It looks great and please see minor comments below.

• In Introduction, the newly cited article by Pardede et al seems irrelevant (ref 15).

• Lines 356-360: the authors mentioned that ‘This finding is not similar to a study done in Ethiopia, which found that the wealth status of the household is not statistically significant for the consumption of foods rich in vitamin A’. It would be valuable if authors elaborate more regarding the inconsistencies.

• Don’t need to cite figures again in discussion. No need to repeat the results – you can highlight the key findings and provide some insights.

• Review again the list of figures and numbering (fig 3 appeared twice in the list and some figures were numbered as figure 1).

• It seems that the editing wasn’t done for final version of the manuscript because there are still some errors in edited parts.

Reviewer #3: (No Response)

7. PLOS authors have the option to publish the peer review history of their article (what does this mean?). If published, this will include your full peer review and any attached files.

Reviewer #2: No

Reviewer #3: No

---

## [Author Response · Author response to Decision Letter 1]

17 May 2021

Response to Reviewers

Spatial distribution and geographical heterogeneity factors associated with poor consumption of foods rich in vitamin A among children aged 6 -23 months in Ethiopia: Geographical weighted regression analysis 

Sofonyas Abebaw Tiruneh1*, Dawit Tefera Fentie2, Seblewongel Tigabu Yigizaw 2, Asnakew Asmamaw Abebe2, Kassahun Alemu Gelaye2.

The authors, extending our great thanks to the editors and reviewers for this manuscript as the stand of this review. The comments raised by the reviewers and editors are vital and defiantly it will improve the quality of the manuscript. Please note that texts and sentences underneath the reviewer’s question and/or comment is the authors' response and reaction to each issue. 

Stay Safe!!!

The Authors.

Reviewer’s comment and question

Reviewer #2: 

Congratulations on this work. It looks greatthe and please see minor comments below.

Noted Thank you

In the introduction, the newly cited article by Pardede et al seems irrelevant (ref 15).

Thank you corrected accordingly.

• Lines 356-360: the authors mentioned that ‘This finding is not similar to a study done in Ethiopia, which found that the wealth status of the household is not statistically significant for the consumption of foods rich in vitamin A’. It would be valuable if authors elaborate more regarding the inconsistencies.

Thank you. 

• Don’t need to cite figures again in discussion. No need to repeat the results – you can highlight the key findings and provide some insights.

Thank you! This might be the study period, sample size, and area difference. 

• Review again the list of figures and numbering (fig 3 appeared twice in the list to and some figures were numbered as figure 1).

Thank you corrected accordingly. 

• It seems that the editing wasn’t done for the final version of the manuscript because there are still some errors in the edited parts.

Corrected. 

Reviewer #3: (No Response)

Thank you for your review.

---

## [Editor Report · Decision Letter 2]

20 May 2021

Spatial distribution and geographical heterogeneity factors associated with poor consumption of foods rich in vitamin A among children age 6 - 23 months in Ethiopia: Geographical weighted regression analysis

PONE-D-20-16369R2

Dear Dr. Tiruneh,

We’re pleased to inform you that your manuscript has been judged scientifically suitable for publication and will be formally accepted for publication once it meets all outstanding technical requirements.

Kind regards,

Andrew Amos Channon, PhD

Academic Editor

PLOS ONE
---

## [Editor Report · Acceptance letter]

24 May 2021

PONE-D-20-16369R2 

</i>Spatial distribution and geographical heterogeneity factors associated with poor consumption of foods rich in vitamin A among children age 6 - 23 months in Ethiopia: Geographical weighted regression analysis</i> 

Dear Dr. Tiruneh:

I'm pleased to inform you that your manuscript has been deemed suitable for publication in PLOS ONE. Congratulations! Your manuscript is now with our production department. 

Kind regards, 

on behalf of

Dr. Andrew Amos Channon 

Academic Editor

PLOS ONE